# The $C_2A$ domain of synaptotagmin is an essential component of the calcium sensor for synaptic transmission

**Matthew R. Bowers, Noreen E. Reist** *

Department of Biomedical Sciences, Molecular, Cellular, Integrative Neurosciences Program, Colorado State University, Fort Collins, CO, United States of America

* reist@colostate.edu

**Data Availability Statement:** All data files associated with this paper are publicly available from the Mountain Scholar database at https://hdl.handle.net/10217/199886 and/or https://doi.org/10.25675/10217/199886.

## Abstract

The synaptic vesicle protein, synaptotagmin, is the principle $Ca^{2+}$ sensor for synaptic transmission. $Ca^{2+}$ influx into active nerve terminals is translated into neurotransmitter release by $Ca^{2+}$ binding to synaptotagmin's tandem C2 domains, triggering the fast, synchronous fusion of multiple synaptic vesicles. Two hydrophobic residues, shown to mediate $Ca^{2+}$-dependent membrane insertion of these C2 domains, are required for this process. Previous research suggested that one of its tandem C2 domains ($C_2B$) is critical for fusion, while the other domain ($C_2A$) plays only a facilitatory role. However, the function of the two hydrophobic residues in $C_2A$ have not been adequately tested *in vivo*. Here we show that these two hydrophobic residues are absolutely required for synaptotagmin to trigger vesicle fusion. Using *in vivo* electrophysiological recording at the *Drosophila* larval neuromuscular junction, we found that mutation of these two key $C_2A$ hydrophobic residues almost completely abolished neurotransmitter release. Significantly, mutation of both hydrophobic residues resulted in more severe deficits than those seen in synaptotagmin null mutants. Thus, we report the most severe phenotype of a $C_2A$ mutation to date, demonstrating that the $C_2A$ domain is absolutely essential for synaptotagmin's function as the electrostatic switch.

## Author summary

The postulated role of synaptotagmin's $C_2A$ domain in triggering neurotransmitter release has fluctuated wildly over the years. Early biochemical experiments suggested that the $C_2A$ domain was essential, while the $C_2B$ domain was superfluous. Then, functional experiments measuring neurotransmitter release *in vivo* following disruptions in $Ca^{2+}$ binding suggested that $C_2B$ was essential, while $C_2A$ was superfluous. Subsequently, the use of more refined mutations to disrupt $Ca^{2+}$ binding indicated that $C_2A$ played a facilitatory role. Here we show two hydrophobic residues of the $C_2A$ domain are *absolutely required* for synaptotagmin-triggered neurotransmitter release. Thus, after over twenty years of research, we now demonstrate that the $C_2A$ domain of synaptotagmin is an essential component of the $Ca^{2+}$ sensor for triggering synaptic transmission *in vivo*.

**Funding:** NER - IOS-1257363 from the National Science Foundation. www.nsf.gov. NER – CRC from the College of Veterinary Medicine and Biomedical Sciences award. https://vetmedbiosci.colostate.edu. The funders had no role in study design, data collection and analysis, decision to publish, or preparation of the manuscript.

**Competing interests:** The authors have declared that no competing interests exist.

## Introduction

Ca$^{2+}$ binding by synaptotagmin triggers the fast, synchronous fusion of maximally-primed synaptic vesicles thereby releasing neurotransmitter onto the postsynaptic cell [1, 2]. In the absence of this Ca$^{2+}$ sensor, evoked release of neurotransmitter is dramatically decreased [3–6]. Synaptotagmin is an integral membrane protein found on synaptic vesicles whose cytosolic domain is composed of two Ca$^{2+}$-binding C2 domains, C$_2$A and C$_2$B [Fig 1A and 1B, [7]]. One end of each C2 domain contains three loops of amino acids, two of which form a Ca$^{2+}$-binding pocket: loops 1 and 3 contain five negatively-charged aspartate residues that coordinate Ca$^{2+}$[8, 9]. In addition, there are two, highly-conserved, hydrophobic residues at the tips of each pocket: one in loop 1, adjacent to the first aspartate residue, and one in loop 3, between the 4$^{th}$ and 5$^{th}$ aspartate residues [Fig 1A,[8, 9]]. Prior to Ca$^{2+}$ influx, the net negative charge of each Ca$^{2+}$-binding pocket results in electrostatic repulsion of the negatively-charged presynaptic membrane, preventing fusion. Upon Ca$^{2+}$ influx, Ca$^{2+}$ binding to the pockets now results in a net positive charge. Accordingly, the electrostatic repulsion of the presynaptic membrane is changed to electrostatic attraction. Thus, synaptotagmin operates as an electrostatic switch [10–13]. Importantly, this electrostatic attraction now brings the hydrophobic residues located at the tips of the pockets into contact with the membrane. Modeling predicts, and *in vitro* studies confirm, that these hydrophobic residues insert into lipid bilayers in a Ca$^{2+}$-dependent manner [14–16], resulting in positive curvature of the membrane that is theorized to promote vesicle fusion [17, 18].

C$_2$A is currently postulated to function as a secondary domain that is merely supportive of C$_2$B, the primary functional domain of synaptotagmin. Importantly, mutation of a hydrophobic tip residue in loop 3 of C$_2$B, which penetrates negatively-charged membranes, is embryonic lethal and causes a decrease in evoked release that is more severe than that seen in *syt$^{null}$* mutants. In comparison, mutating the analogous residue in the C$_2$A domain does not impact viability and only inhibited neurotransmitter release by 50% [19]. However, the functional impact of mutations of the C$_2$A hydrophobic residue in loop 1 has not been studied *in vivo*, nor has the impact of tandem mutations in both loops 1 and 3 simultaneously.

Since replacing the loop 3 hydrophobic residue in C$_2$B with a large, polar glutamate prevented phospholipid-binding in vitro [19], we made homologous mutations of the loop 1 and loop 3 residues in C$_2$A (Fig 1D). Electrophysiological recordings at the *Drosophila* neuromuscular junction revealed that mutation at either the loop 1 (this report) or loop 3 site [19] in isolation resulted in an ~50% reduction in evoked transmitter release, again suggesting C$_2$A plays only a facilitatory role. Surprisingly, mutation of both the loop 1 and loop 3 sites simultaneously resulted in an almost complete abolishment of evoked release. This reduction in transmission in the tandem mutation was actually more severe than that observed in synaptotagmin null mutants. The current study establishes that these two hydrophobic residues of C$_2$A, which have been shown to mediate Ca$^{2+}$-dependent effector interactions *in vitro* [17, 18, 20, 21], are absolutely required for evoked transmitter release *in vivo*. Thus, we report the most severe deficits caused by any C$_2$A domain mutation to date and demonstrate that the C$_2$A domain is an essential component for translating synaptotagmin's electrostatic switch function into vesicle fusion with the presynaptic membrane.

## Materials and methods

### *Drosophila* lines

We generated mutants with hydrophobic to hydrophilic substitutions of two residues. A transgenic wild type (*P[sytWT]*) was used as a positive control across all experiments. For a direct

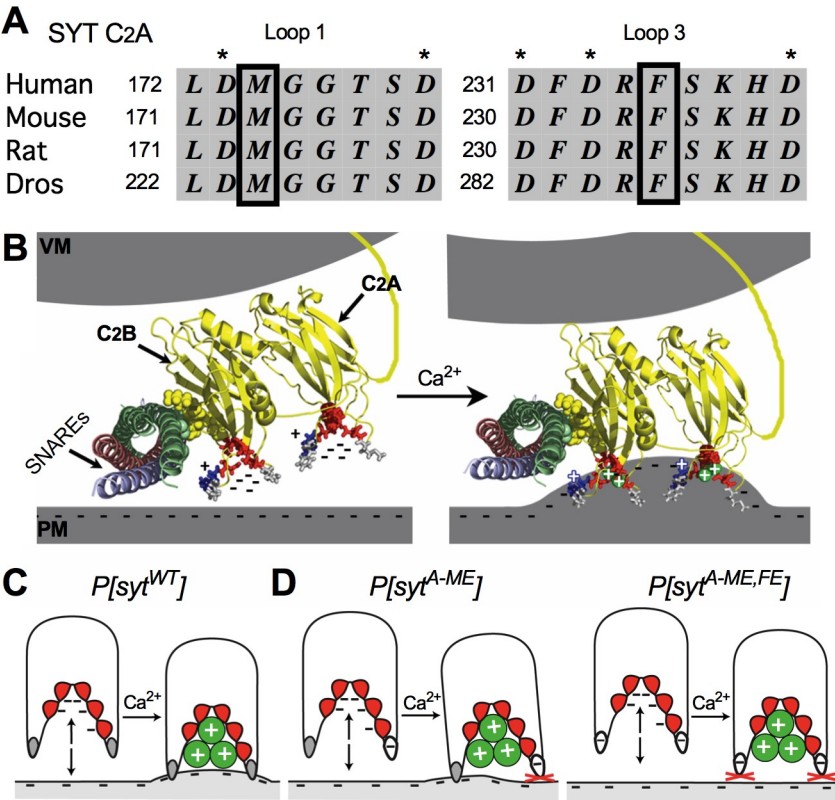

**Fig 1. Synaptotagmin structure and C$_2$A mutations. A,** Protein alignment of loops 1 and 3 of the C$_2$A domain of synaptotagmin 1 from Human, Mouse, Rat, and *Drosophila* (* = Ca$^{2+}$ binding aspartates, boxes = loop 1 and loop 3 hydrophobic tip residues) **B,** Crystal structure of synaptotagmin and the SNARE complex showing a postulated role of the C2 domains in triggering fusion, adapted from [19]. Negatively charged residues of the Ca$^{2+}$ binding pockets are shown as sticks in red, the hydrophobic residues at the tips of these pockets are shown as sticks in gray, and Ca$^{2+}$ ions are shown as green spheres. VM = vesicle membrane and PM = presynaptic membrane. **C,** A cartoon depiction of the C$_2$A domain. Colors as in panel B. **D,** Hydrophilic glutamic acid substitutions are indicated in white. Sequential mutation of C$_2$A hydrophobic tip residues to hydrophilic residues is predicted to increasingly disrupt synaptotagmin's ability to penetrate, warp and disorder lipids of the presynaptic membrane.

comparison of the level of evoked transmitter release in the most severe mutation, a previously characterized synaptotagmin null line (*syt$^{null}$*) was used as a negative control [22]. Using the *Drosophila syt1* coding sequence [7, 23], a wild type control, a M224E, and a M224E/F286E mutant cDNA were synthesized by GeneWiz (South Plainfield, New Jersey) (Fig 1A and 1D). The cDNA was flanked by unique 5' EcoRI and 3' BglII restriction sites for directional sub-cloning into the pUAST-attB vector to place them under the control of the UAS promoter. The transgenes were injected into *Drosophila* embryos by BestGene (Chino Hills, California) where they were inserted into the attP2 landing site on the third chromosome using the PhiC31 targeted insertion system [24]. These *syt 1* transgenes were driven pan-neuronally by the UAS/Gal4 system [25] using the *elav* promoter [26]. All transgenes were expressed in the absence of endogenous synaptotagmin 1 by crossing them into a synaptotagmin 1 null mutant background, *syt$^{AD4}$* [22, 23]. As no sex selection was employed, both males and females were used across all experiments. This study used the following genotypes: *yw; sytAD4elavGal4/ sytAD4; P[UASsyt1WT]/+* (referred to as *P[sytWT]* or control), *yw; sytAD4elavGal4/ sytAD4; P[UASsyt1C2A-M224E]/+* (referred to as *P[sytA-ME]*), *yw; sytAD4elavGal4/ sytAD4; P*

*[UASsyt1 C2A-M224E,F286E]/+* (referred to as *P[sytA-ME,FE]*) and *yw; sytAD4elavGal4/ sytAD4* (referred to as *syt^null^*). All experiments used 3$^{rd}$ instar larvae (L3).

## Solutions

HL3.1 saline [70 mM NaCl, 5 mM KCl, 4 mM MgCl$_2$, 10 mM NaHCO$_3$, 5 mM Trehalose, 115 mM sucrose, 5 mM HEPES, pH 7.2 [27]], with the indicated Ca$^{2+}$ concentrations, was used in all experiments. Phosphate buffered saline (PBS) consisted of [137 mM NaCl, 1.5 mM KH$_2$PO$_4$, 2.7 mM KCl, 8.1 mM Na$_2$HPO$_4$].

## Immunoblotting

Synaptotagmin expression levels were determined using western blot analysis with actin levels serving as a loading control. The CNSs of L3s were dissected in HL3.1 saline where the Ca$^{2+}$ was omitted to decrease vesicle fusion events during dissection. Individual CNSs were placed in Laemmli buffer (Bio-Rad, Hercules, CA) containing 5% β-mercaptoethanol, sonified with five 0.3 sec pulses at 1 Hz using a Branson Sonifier 450 (VWR Scientific, Winchester, PA), and separated by SDS-PAGE with 15% acrylamide. They were then transferred to Immobilon membranes (Millipore, Bedford, MA), and washed in blocking solution [5% milk, 4% normal goat serum (NGS, Fitzgerald Industries International, Acton, MA), 1% bovine serum albumin (BSA, Millipore-Sigma, Burlington, MA), and 0.02% NaN$_3$ in PBS containing 0.05% Tween 20 (PBS-Tween, Fisher BioReagents, Fair Lawn, NJ)]. The membranes were then incubated over-night at 4°C with a 1:2,500 dilution of anti-synaptotagmin antibody, Dsyt-CL1 [2] and 1:10,000 dilution of anti-actin antibody, (MAB 1501, Millipore Bioscience Research Reagents, Billerica, MA) in PBS-Tween containing 10% NGS and 0.02% NaN$_3$, washed in PBS-Tween for 1–3 hours, and probed with secondary antibodies at a 1:5,000 dilution of peroxidase-conju-gated AffiniPure Goat Anti-Rabbit IgG (Jackson ImmunoResearch, West Grove, PA) and a 1:5,000 dilution of peroxidase-conjugated AffiniPure Donkey Anti-Mouse IgG (Jackson ImmunoResearch, West Grove, PA) in PBS-Tween containing 10% NGS for 1 hour at room temperature, and washed in PBS-Tween for 30 min. Protein bands were visualized on an Epi-chemi$^3$ Darkroom with Labworks Imaging Software (UVP BioImaging, Upland, CA). To quantify expression levels within each blot, synaptotagmin/actin ratios were calculated and normalized to the mean synaptotagmin/actin ratio of the transgenic WT control lanes. This permitted comparison of synaptotagmin expression levels between blots. Outliers in loading amount, based on actin levels, were excluded from analysis. The analysis included at least 11 individual CNSs per genotype.

## Immunolabeling

The localization of the transgenic synaptotagmin protein was visualized by immunohis-tochemistry. L3s were dissected in Ca$^{2+}$-free HL3.1, fixed in PBS containing 4% formaldehyde for 1 hour, incubated with a 1:400 dilution of Dsyt-CL1 in dilution media [PBS with 0.1% Tri-ton (PBST), 1% BSA, and 1% NGS] overnight at 4°C, washed in PBST for 1–3 hours, incubated in dilution media containing a 1:400 dilution of Alexa Fluor 488 goat anti-rabbit antibody (Invitrogen, Carlsbad, CA) for 1 hour at room temperature, washed in PBST for one hour, and mounted on microscope slides in Citifluor (Ted Pella, Redding, CA). Confocal images of the neuromuscular junction on muscle fibers 6 and 7 were taken on a Zeiss 880 light-scanning microscope (Zeiss, White Plains, NY), with a 40x objective and Zeiss Zen 2.1 acquisition soft-ware, version 11.0.3.190.

## Electrophysiological recording and analyses

Electrophysiological recordings were made with an Axoclamp 2B amplifier (Molecular Devices, Sunnyvale, CA), a Powerlab 4/30 A/D converter (ADInstruments, Sydney, Australia), using LabChart software (ADInstruments, Sydney, Australia). L3s were dissected in Ca$^{2+}$-free HL3.1 saline to expose the body wall musculature and the CNSs were removed. The saline was then changed to HL3.1 with 1mM Ca$^{2+}$. Intracellular recordings were made from muscle fiber 6 of abdominal segments 3 and 4 using 10–20 MΩ intracellular electrodes that were pulled using a Sutter model P-97 micropipette puller (Novato, CA) and filled with 3 parts 2 M K$_3$C$_6$H$_5$O$_7$ to 1 part 3 M KCl. The resting potential was held at -65 mV by applying no more than ±1 nA of current. The nerve fiber was stimulated using an A360 stimulus isolator (World Precision Instruments, Sarasota, FL) through a glass suction electrode filled with HL3.1 containing 1mM Ca$^{2+}$ and broken to have an ~1 micron tip.

**Evoked release.**   Ten excitatory junction potentials (EJPs) were stimulated at 0.04 Hz and averaged for each muscle fiber. Mean responses are reported for 12–14 muscle fibers per genotype.

**Spontaneous release.**   Spontaneous miniature EJPs (mEJPs) were recorded for 3 min prior to any external stimulation. Recordings were blinded and randomized and mEJPs were identified manually. mEJP frequency was determined by counting the number of events that occurred during the second minute of recording. The average amplitude of the first 100 of this population of mEJPs was also calculated. Mean responses are reported for 12–14 muscle fibers per genotype.

**Paired pulse.**   For each muscle fiber, the nerve was stimulated with pairs of pulses having interpulse intervals of 10ms, 20ms, 50ms, and 100ms. For each interpulse interval, five pairs of pulses separated by 5 sec were averaged. Each interpulse interval test was also separated by 5 sec. The amplitude of the first EJP was calculated from the baseline to the first peak. The amplitude of the second EJP was calculated from the trough following the first EJP to the peak of the second EJP. The amplitude of the second EJP was divided by the amplitude of the first EJP to yield a paired pulse ratio (PPR). The mean PPRs of 12–14 fibers per genotype are reported.

**Ca$^{2+}$ curves.**   Five EJPs evoked at 0.5 Hz were recorded from an individual muscle fiber across at least 3 different Ca$^{2+}$ concentrations between 0.05 mM and 5 mM (0.05 mM, 0.25 mM, 0.5 mM, 1 mM, 1.5 mM, 2.5 mM, 5 mM) and averaged, yielding a mean EJP amplitude at each Ca$^{2+}$ level. The first 5 EJPs were always recorded in HL3.1 with 1.5 mM Ca$^{2+}$ and, following stimulation in at least 2 other Ca$^{2+}$ levels, 5 more EJPs were recorded in HL3.1 with 1.5 mM Ca$^{2+}$. Recordings were only considered for analysis if the mean amplitude of the final EJPs in 1.5 mM Ca$^{2+}$ was $\geq$ 90% of the initial mean EJP amplitude. Mean responses are reported for at least 12 muscle fibers per Ca$^{2+}$ concentration. Lines of best fit were calculated using a nonlinear regression analysis, which provided EC50s and their 95% confidence intervals for each genotype. Each response level was normalized to the maximum value predicted by the line of best fit. Additionally, the hillslope for each curve was calculated to compare cooperativity of Ca$^{2+}$-dependent release.

## Experimental design and statistical analysis

Statistical analyses were performed using Prism 8 (GraphPad software, La Jolla, CA). All datasets included a minimum of 11 samples per genotype. In all electrophysiological experiments, recordings of mutants and controls were interspersed. Direct comparisons were only made between recordings done within the same time period, as absolute responses can be impacted by minor variations in solutions. All experiments requiring manual analysis of events had the genotypes blinded to the researcher. For comparison of two genotypes with Gaussian

distributions of their datasets, we used unpaired student's t-tests. For comparisons of three genotypes with Gaussian distribution of their datasets, we used one-way ANOVAs with Tukey post hoc tests of multiple comparisons to determine significance between all three genotypes. In the paired pulse experiments, data were analyzed with a repeated measures two-way ANOVA with Tukey post hoc tests. When datasets showed non-gaussian distributions, Kruskal-Wallis tests were used to compare the 3 genotypes, with Dunn's post hoc tests of multiple comparisons. An alpha p-value of 0.05 was considered significant for all of the above tests. To compare Ca$^{2+}$ curves, a line of best fit was determined using a nonlinear regression model and the 95% confidence intervals of the EC50 of each genotype were compared. If the confidence intervals didn't overlap, the genotypes were considered significantly different.

## Results

### Synaptotagmin transgenes

*P[sytWT]*, *P[sytA-ME]*, and *P[sytA-ME,FE]*.  Mutation of the hydrophobic residue at the tip of loop 3 of the C$_2$A Ca$^{2+}$-binding pocket inhibits evoked release by 50% [19], yet the *in vivo* function of the hydrophobic residue at the tip of loop 1 is unknown (see Fig 1A, Loop 1, M). Since both of these hydrophobic residues have been shown to mediate Ca$^{2+}$-dependent interactions *in vitro* [16, 18, 20], we tested whether the loop 1 hydrophobic residue is required for efficient neurotransmitter release. We generated two lines containing mutations of this loop 1 methionine. In the first, only the loop 1 methionine was mutated to a hydrophilic glutamic acid (Fig 1D, *P[sytA-ME]*). In the second, both the loop 1 and loop 3 hydrophobic tip residues of C$_2$A were mutated to glutamic acids (Fig 1D, *P[sytA-ME,FE]*). In all experiments, transgenic synaptotagmin was expressed in the *syt$^{null}$* background such that the only source of synaptotagmin 1 was from the transgene [22].

### C$_2$A hydrophobic residues are required for synaptotagmin function

Ca$^{2+}$-evoked neurotransmitter release in the single *P[sytA-ME]* mutant was decreased to a similar extent as that seen previously in the single *P[sytA-FE]* mutant [19]. Electrophysiological recording of excitatory junction potentials (EJPs) from larval muscle fibers revealed an ~50% decrease in EJP amplitude in *P[sytA-ME]* compared to *P[sytWT]* (Fig 2A and 2B). EJP amplitude in *P[sytWT]* was 30.8 ± 1.8 mV (mean ± SEM, n = 12). Whereas in *P[sytA-ME]*, it was significantly reduced at only 14.6 ± 1.3 mV (mean ± SEM, n = 14; one-way ANOVA F(2,37) = 137.7, p < 0.0001, Tukey post hoc p < 0.0001). This partial block of fast, synchronous neurotransmitter release is consistent with the idea that C$_2$A plays only a facilitatory role in synaptotagmin function [19, 28, 29].

Surprisingly, the double mutant, *P[syt$^{A-ME,FE}$]*, nearly abolished Ca$^{2+}$-evoked neurotransmitter release (Fig 2A and 2B). EJP amplitude in *P[syt$^{A-ME,FE}$]* mutants was only 1.2 ± 0.2 mV (mean ± SEM, n = 14). Since the double mutant showed such a dramatic decrease in EJP amplitude compared to both *P[syt$^{WT}$]* and *P[syt$^{A-ME}$]* (one-way ANOVA F(2,37) = 137.7, p < 0.0001, Tukey post hoc p < 0.0001), we also compared evoked transmitter release in *P[syt$^{A-ME,FE}$]* mutants and *syt$^{null}$* larvae, which express no synaptotagmin [22]. Importantly, we found that the *P[syt$^{A-ME,FE}$]* double mutant had a significantly decreased EJP amplitude even when compared to *syt$^{null}$* larvae (Fig 2C and 2D; mean ± SEM: *P[syt$^{A-ME,FE}$]* = 2.5 ± 0.4 mV, n = 21 and *syt$^{null}$* = 5 ± 0.8 mV, n = 21; unpaired t-test t(40) = 2.896, p = 0.0061). Thus, residues required for Ca$^{2+}$-dependent membrane penetration by both C2 domains are absolutely required for synaptotagmin to function as the Ca$^{2+}$ sensor for fast, synchronous neurotransmitter release. While we've known for many years the critical nature of the C$_2$B domain for

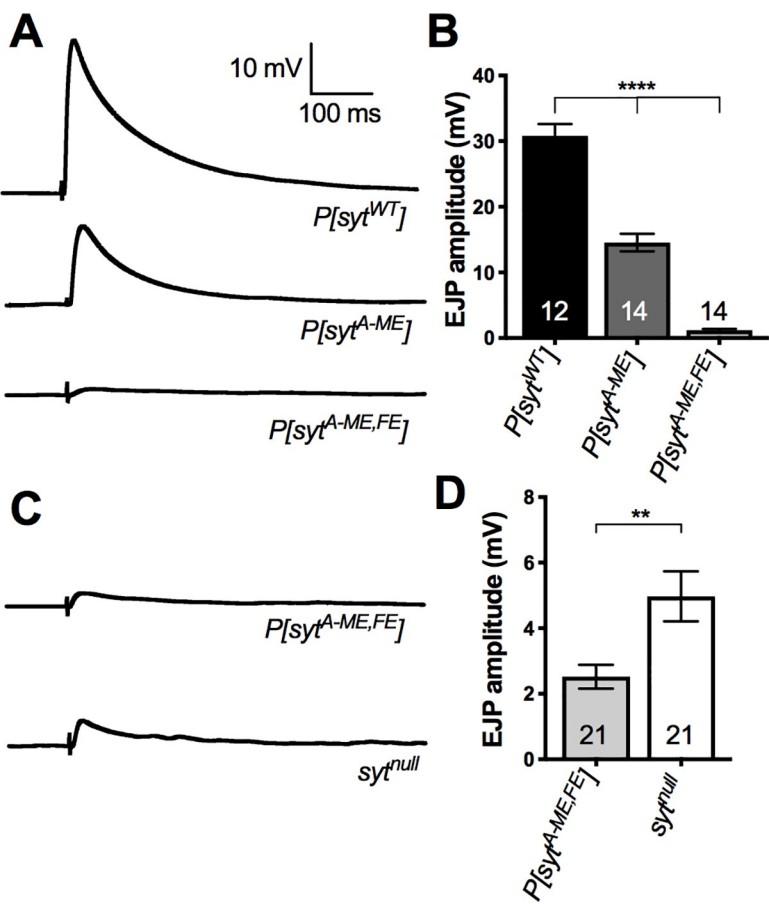

**Fig 2. Mutation of the hydrophobic tip residues disrupts evoked transmitter release.** The single hydrophobic mutation decreased neurotransmitter release by 50% while the double mutation inhibited release to a greater extent than that seen in $syt^{null}$ mutants. **A,** Representative traces of EJPs for $P[syt^{WT}]$, $P[syt^{A-ME}]$, and $P[syt^{A-ME,FE}]$. **B,** Mean EJP amplitude ± SEM for $P[syt^{WT}]$, $P[syt^{A-ME}]$, and $P[syt^{A-ME,FE}]$ (Tukey multiple comparisons, p < 0.0001 = $^{****}$). **C,** Representative traces of EJPs for $P[syt^{A-ME,FE}]$ and $syt^{null}$. **D,** Mean EJP amplitude ± SEM for $P[syt^{A-ME,FE}]$ and $syt^{null}$ (Tukey multiple comparisons, p < 0.01 = $^{**}$).

triggering neurotransmitter release [2, 4, 19, 30, 31], no previous $C_2A$ *mutation* has resulted in synaptic deficits more severe than those in $syt^{null}$ mutants *in vivo*.

### Expression and localization of transgenic synaptotagmin

The deficits in evoked release are not a result of mis-expression or mis-localization of the transgenic proteins. Western analysis was performed on single CNSs from larvae using our anti-synaptotagmin antibody [DsytCL1, [2]] and an anti-actin antibody (MAB 1501) as a loading control. Synaptotagmin expression in $P[syt^{A-ME}]$ and in $P[syt^{A-ME,FE}]$ was 114.4 ± 24.0% and 78.7 ± 17.6% (respectively, mean ± SEM, n = 13 and n = 11) of that in $P[syt^{WT}]$ (n = 13). There were no significant differences in transgenic synaptotagmin expression levels between the control and either mutant line (Fig 3A and 3B, one-way ANOVA F(2,34) = 0.8165, p = 0.4505). Immunohistochemical labeling of the larval body wall musculature with anti-synaptotagmin antibody was used as a non-quantitative measure of protein localization, which demonstrated that the transgenic synaptotagmin was highly concentrated in synaptic boutons at the neuromuscular junction in both transgenic mutants and the transgenic WT control (Fig 3C). Thus, the transgenic protein was appropriately targeted to synaptic sites.

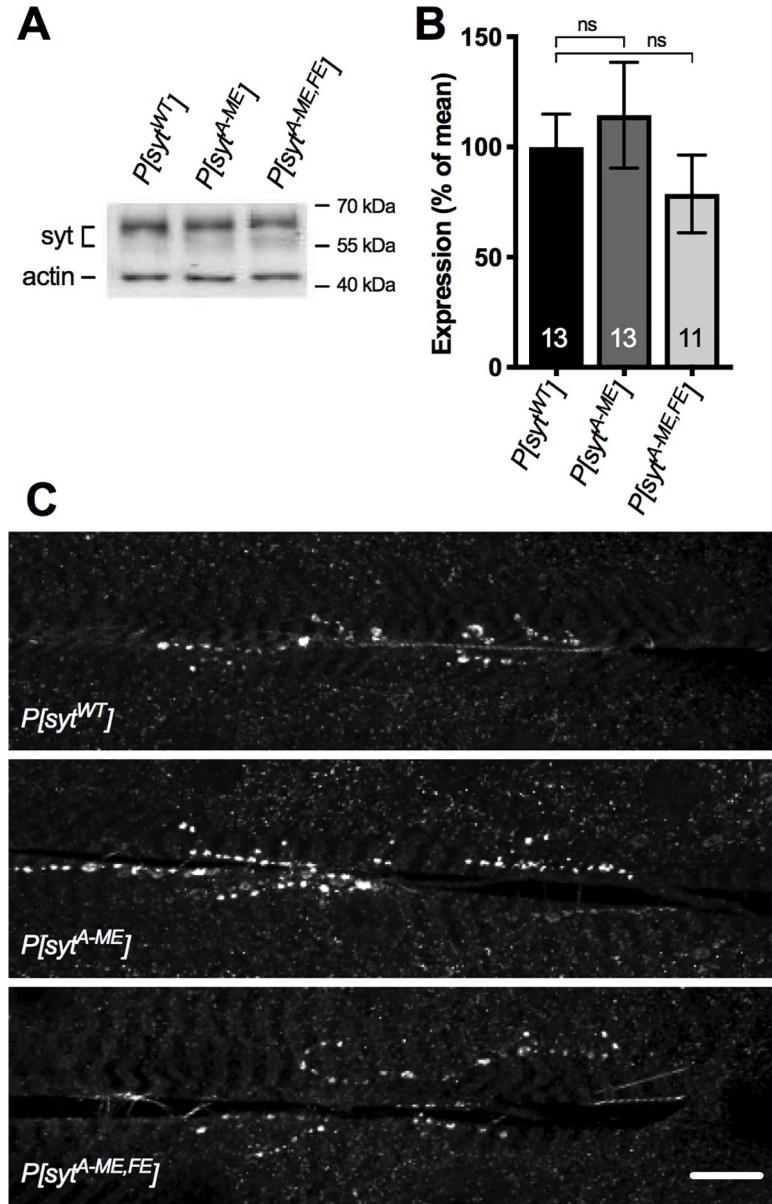

**Fig 3. Expression and localization of synaptotagmin are unaffected by hydrophobic mutations. A,** Representative western blots of transgenic synaptotagmin expression levels with actin as a loading control. **B,** Mean protein expression levels ± SEM, normalized to actin (One-way ANOVA, no significant differences). **C,** Representative confocal images of larval neuromuscular junctions labeled with anti-synaptotagmin antibodies (scale bar = 20 μm). Synaptotagmin is appropriately concentrated at synaptic sites in all three genotypes.

## C$_2$A hydrophobic mutations result in decreased release probability

A decrease in evoked transmitter release could result from a decrease in vesicular loading of neurotransmitter which would decrease quantal size and/or a decrease in release probability. The amplitude of spontaneous single vesicle fusion events, mEJPs, at the neuromuscular junction provides a convenient indication of quantal size, assuming there's no disruption in post-synaptic responsiveness. The amplitude of these single vesicle fusion events was unchanged between the mutants and the control (Fig 4A and 4B; mean ± SEM: *P[syt$^{WT}$]* = 1.1 ± 0.06 mV

n = 20, $P[syt^{A-ME}]$ = 1.2 ± 0.08 mV n = 16, $P[syt^{A-ME,FE}]$ = 0.9 ± 0.05 mV n = 18; Kruskal-Wallis test H(2) = 7.61, p = 0.0222, Dunn's multiple comparisons test, $P[syt^{WT}]$ vs $P[syt^{A-ME}]$ p > 0.9999, $P[syt^{WT}]$ vs $P[syt^{A-ME,FE}]$ p = 0.07). The lack of any difference between the control and either mutant is consistent with both vesicular loading and postsynaptic responsiveness being unchanged. Several synaptotagmin mutations result in an increase in the frequency of spontaneous fusion events [5, 32]. Therefore, we analyzed mEJP frequency and found no change between the mutants and the control (Fig 4A and 4C; mean ± SEM: $P[syt^{WT}]$ = 3.9 ± 0.5 Hz n = 20, $P[syt^{A-ME}]$ = 3.7 ± 0.3 Hz n = 16, $P[syt^{A-ME,FE}]$ = 3.9 ± 0.3 Hz n = 18; one-way ANOVA F(2, 51) = 0.1247, p = 0.8806). Thus, neither hydrophobic mutation caused a significant change in either the quantal size or the rate of spontaneous fusion events.

While changes in quantal size cannot account for the differences in EJP amplitude we observed, another possibility is that the hydrophobic mutations decrease the probability of release. Since the ratio of two paired pulses (paired pulse ratio, PPR) is inversely correlated to release probability [33], we conducted a paired pulse analysis. The PPRs for $P[syt^{A-ME}]$ and $P[syt^{A-ME,FE}]$ were significantly different from control across all interpulse intervals (mean ± SEM. $P[syt^{WT}]$ n = 13: 10 ms = 0.6 ± 0.05, 20 ms = 0.7 ± 0.04, 50 ms = 0.9 ± 0.03, 100 ms = 0.9 ± 0.01. $P[syt^{A-ME}]$ n = 14: 10 ms = 1.1 ± 0.1, 20 ms = 1.0 ± 0.08, 50 ms = 1.1 ± 0.06, 100 ms = 1.1 ± 0.04. $P[syt^{A-ME,FE}]$ n = 12: 10 ms = 2.7 ± 0.3, 20 ms = 1.8 ± 0.2, 50 ms = 1.6 ± 0.1, 100 ms = 1.5 ± 0.2. Two-way repeated measures ANOVA, F (36, 108) = 2.190 p = 0.001; Tukey post hoc–$P[syt^{WT}]$ vs $P[syt^{A-ME}]$: 10 ms p = 0.0005, 20 ms p = 0.01, 50 ms p = 0.007, 100 ms p = 0.0003. $P[syt^{WT}]$ vs $P[syt^{A-ME,FE}]$: 10 ms p < 0.0001, 20 ms p = 0.0001, 50 ms p = 0.0008, 100 ms p = 0.04). Furthermore, the PPR of the double mutant was significantly greater than that of the single mutant for every interpulse interval, except 100 ms (Fig 5A and 5B $P[syt^{A-ME}]$ vs $P[syt^{A-ME,FE}]$, Tukey post hoc–$P[syt^{A-ME}]$ vs $P[syt^{A-ME,FE}]$: 10 ms p = 0.0003, 20 ms p = 0.003, 50 ms p = 0.01, 100 ms p = 0.2). These results indicate that a decreased release probability could account for the decrease in neurotransmitter release, which is further bolstered by the finding that the release probability scales inversely with the severity of the mutation.

## C$_2$A hydrophobic mutations decrease the apparent Ca$^{2+}$ affinity of release

Fast, synchronous neurotransmitter release is a Ca$^{2+}$-dependent, cooperative process [34]. In order to test whether the C$_2$A hydrophobic mutations affect the Ca$^{2+}$ dependence of synaptotagmin-triggered release, we recorded EJPs at a variety of extracellular Ca$^{2+}$ concentrations, ranging from 0.05 to 5 mM. In our dose-response curve, the intermediate decrease of EJP amplitude in $P[syt^{A-ME}]$ mutants and the dramatic decrease of EJP amplitude in $P[syt^{A-ME,FE}]$ mutants compared to control are evident at all Ca$^{2+}$ concentrations (Fig 6A). We used nonlinear regression to fit curves to the data (Fig 6A solid curves). To determine whether these hydrophobic mutations impact the cooperativity of release, we compared the Hill coefficient, calculated from the equations of the curve for each genotype, and found no significant differences (Hill slope: $P[syt^{WT}]$ = 1.8, $P[syt^{A-ME}]$ = 1.8, $P[syt^{A-ME,FE}]$ = 1.8). Thus, mutations of these C$_2$A hydrophobic residues do not impact the cooperativity of Ca$^{2+}$-dependent neurotransmitter release.

To assess the apparent Ca$^{2+}$ affinity of release, the response at each Ca$^{2+}$ level was normalized to the maximal response predicted by the non-linear regression equation for each genotype. Consistent with the decrease in Ca$^{2+}$ affinity for phospholipid binding mutants reported previously [35], Fig 6B displays a similar shift in the curves for the Ca$^{2+}$ dependence of neurotransmitter release. (EC50, 95%CI: $P[syt^{WT}]$ = 0.6 mM, 0.5–0.7 mM, n = 13; $P[syt^{A-ME}]$ = 0.9 mM, 0.8–1.0 mM, n = 12; $P[syt^{A-ME,FE}]$ = 1.0 mM, 0.8–1.2 mM, n = 12; non-overlapping confidence intervals compared to control). The rightward shift of the Ca$^{2+}$ curves demonstrates that

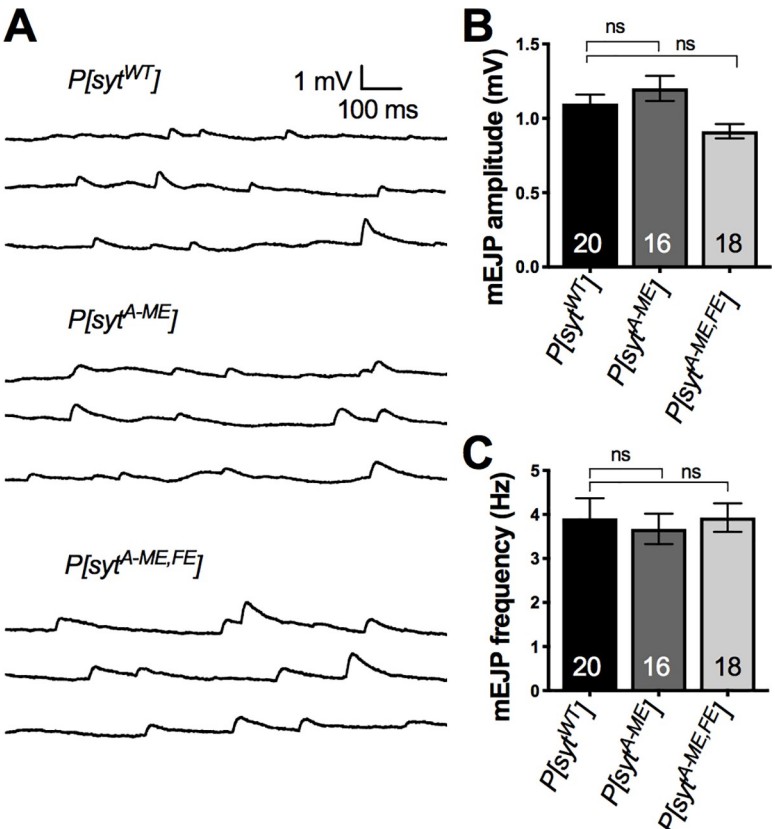

**Fig 4. Spontaneous events are unaffected by hydrophobic mutations. A,** Representative traces of mEJPs for *P[syt^WT^]*, *P[syt^A-ME^]*, and *P[syt^A-ME,FE^]*. **B,** Mean mEJP amplitudes ± SEM (Kruskal Wallis test with Dunn's multiple comparison test, no significance = ns). **C,** Mean mEJP frequency ± SEM (Kruskal Wallis test with Dunn's multiple comparison test, no significance = ns).

both of the C$_2$A hydrophobic mutants caused a decrease in the apparent Ca$^{2+}$ affinity of release.

## Discussion

We investigated the role of two hydrophobic residues in the C$_2$A domain of synaptotagmin in neurotransmitter release. Mutation of the C$_2$A loop 1 hydrophobic residue (*syt^A-ME^*) resulted in a 50% reduction of evoked release (Fig 2B), consistent with previous findings for the loop 3 hydrophobic residue (*syt^A-FE^*, [19]). Notably, mutation of both of these hydrophobic residues in tandem (*syt^A-ME,FE^*) nearly abolished the evoked response (Fig 2B). These deficits could result from the decreased release probability (Fig 5). Analysis of spontaneous release suggests that neither vesicle loading (mEJP amplitude) nor frequency of events (mEJP frequency) played a role in the observed deficits (Fig 4). Evoked responses at varying Ca$^{2+}$ levels revealed that the apparent Ca$^{2+}$ affinity of release was decreased by either our single or double hydrophobic residue mutations in C$_2$A, but the Ca$^{2+}$ cooperativity of release was unaffected (Fig 6). Importantly, the double mutation decreased evoked release significantly more than the complete absence of synaptotagmin 1 (Fig 2D), making it the most severe mutation of the C$_2$A domain of synaptotagmin to date.

Previous *in vivo* and in culture analyses of C$_2$A and C$_2$B domain function suggested that the C$_2$B domain was essential, while C$_2$A played a secondary role. C$_2$B mutations resulted in

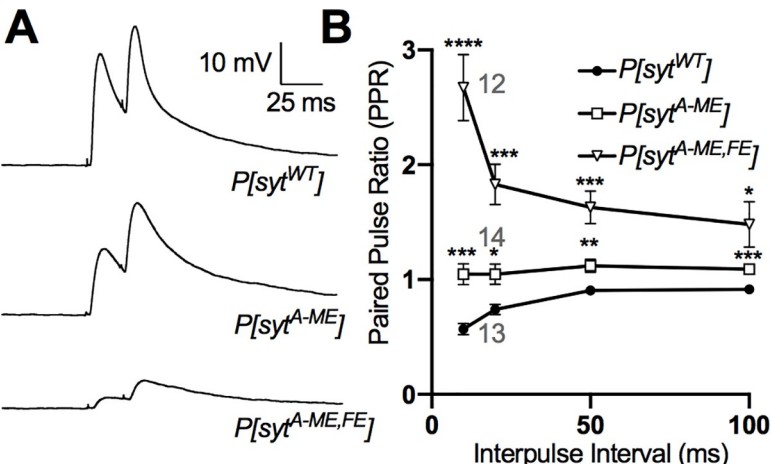

**Fig 5. Probability of release is decreased by hydrophobic mutations.** Probability of release was determined using a paired pulse protocol with interpulse intervals of 10, 20, 50, and 100ms. **A,** Representative traces of paired EJPs for $P[syt^{WT}]$, $P[syt^{A-ME}]$, and $P[syt^{A-ME,FE}]$ with a 20ms interpulse interval. **B,** Mean paired pulse ratios ± SEM for $P[syt^{WT}]$, $P[syt^{A-ME}]$, and $P[syt^{A-ME,FE}]$ (Two-way Repeated Measures ANOVA with Tukey post-hoc, $p < 0.05$ = *, $p < 0.01$ = **, $p < 0.001$ = ***, $p < 0.0001$ = ****). Indicated differences are between mutants and $P[syt^{WT}]$, though the paired pulse ratio was significantly different ($p < 0.05$) between $P[syt^{A-ME}]$, and $P[syt^{A-ME,FE}]$, for all interpulse intervals except 100ms.

dominant negative effects and lethality, while C₂A mutations were viable and only decreased release by a maximum of 50–80%. Specifically, mutations disrupting $Ca^{2+}$ binding resulted in: a dominant-negative effect or lethality in C₂B [2, 30], but only a 0–80% decrease in evoked release in C₂A [10, 28, 29, 36]. Mutations that altered the polylysine motif resulted in: an ~40–50% decrease in evoked transmitter release in C₂B [31, 37–39], yet did not impair evoked release in C₂A [40]. Similarly, mutation of a loop 3 positively-charged residue involved in electrostatic interactions with membranes resulted in: a 60–80% decrease in evoked release in C₂B [41, 42], see however [39], and only an ~45–55% decrease in C₂A [39, 41–43]. Importantly, mutation of the loop 3 hydrophobic residue resulted in: embryonic lethality in C₂B, but only a 50% decrease in evoked release in C₂A [19]. Taken together, these results led to the understanding that C₂B was critical, while C₂A only acted in a facilitatory manner.

The current study challenges this longstanding idea regarding the significance of the C₂A domain. The lethality caused by mutation of the C₂B loop 3 hydrophobic residue is still the most severe synaptotagmin phenotype to date [19] and demonstrates the predominant role of the C₂B domain *in vivo*. Here we report that simultaneous mutation of both the loop 1 and loop 3 hydrophobic tip residues to negatively-charged glutamates in C₂A resulted in the most dramatic deficit ever observed for a C₂A domain mutation. This could be the result of either removal of hydrophobicity or increased electrostatic repulsion of the negatively charged membrane. However, previous work found that mutation of the loop 3 hydrophobic tip residue in C₂A to *either* a hydrophilic, non-charged tyrosine *or* a hydrophilic, negatively-charged glutamate had an equal inhibitory impact on evoked transmitter release. Both decreased release by ~50% [19]. Thus, the net charge at the loop 3 hydrophobic site was inconsequential. Rather, the hydrophilic nature of the mutation correlated with the disruption of function [19]. As such, we chose to use glutamate substitutions at both hydrophobic sites in C₂A for the current study. While the C₂A double glutamate mutant is still viable, this is the first C₂A mutation to result in less neurotransmitter release than that observed in $syt^{null}$ larvae. Thus, the hydrophobic tip residues of *both* domains are essential for synaptotagmin-triggered vesicle fusion. This demonstrates for the first time that C₂A plays more than a facilitatory role; it is absolutely

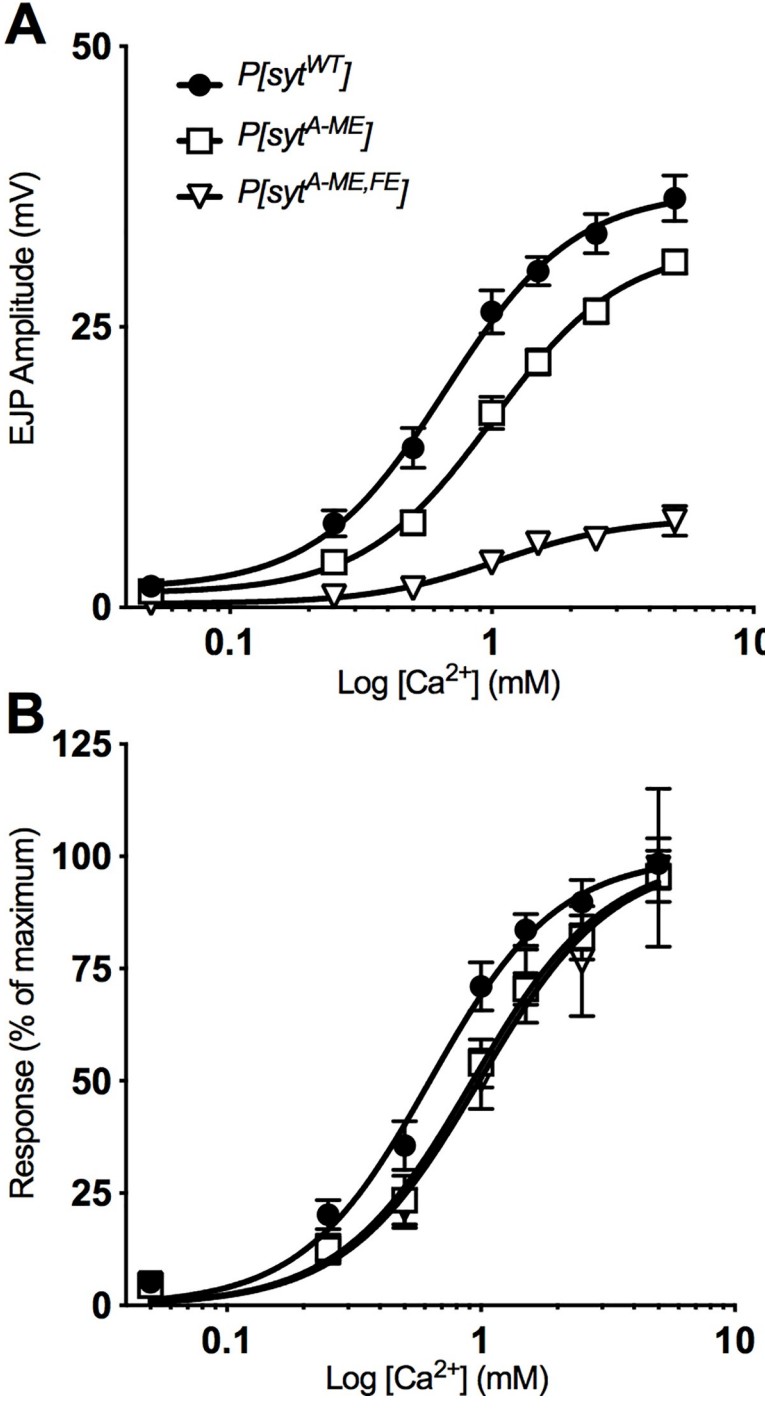

**Fig 6. The apparent Ca$^{2+}$ affinity of release is decreased by hydrophobic mutations. A,** Mean EJP amplitude ± SEM across a range of Ca$^{2+}$ concentrations fit with a nonlinear regression. **B,** Ca$^{2+}$ curves normalized to maximum EJP amplitude predicted by the nonlinear regression. The significant rightward shift in the curve (EC50, non overlapping confidence intervals) indicates a decrease in the apparent Ca$^{2+}$ affinity of neurotransmitter release.

required for synaptotagmin to function as the Ca$^{2+}$ sensor for fast, synchronous neurotransmitter release.

The mechanism(s) by which these hydrophobic residues exert their effects has been extensively studied *in vitro*. These include SNARE interactions with, and membrane penetration by, synaptotagmin. The impact of loop 1 and loop 3 hydrophobic residue mutations on SNARE interactions are controversial, with some studies indicating mutation of the hydrophobic residues has an impact on SNARE binding [18, 19] while others indicate that there is no impact on SNARE binding [21]. However, even the studies reporting decreased t-SNARE binding in mutants with decreased hydrophobicity demonstrated that the changes in synaptotagmin's membrane interactions, and NOT the changes in t-SNARE binding, correlated with the changes in synaptotagmin function *in vivo* [19]. Furthermore, the experiments that showed decreased Ca$^{2+}$-dependent interactions between t-SNAREs and hydrophobic synaptotagmin mutants required membrane-embedded t-SNAREs [18, 19]. Thus, the apparent decrease in Ca$^{2+}$-dependent SNARE interactions may actually be a result of disrupting membrane interactions.

There is, however, ample evidence supporting a role for the hydrophobic tip residues in membrane penetration. The loop 1 and loop 3 hydrophobic residues of both C2 domains insert into membranes following Ca$^{2+}$ binding, implicating membrane penetration, and resultant membrane bending and lipid disorder, as important downstream effector interactions of Ca$^{2+}$ binding [15, 20]. *In vitro* membrane tubulation and liposome fusion assays required Ca$^{2+}$-dependent synaptotagmin C2 domain insertion into membranes [15, 18]. Mutations that prevented membrane penetration blocked liposome fusion and tubulation [17]. In cultured PC12 cells, these same mutations prevented vesicular cargo release [21]. Mutations that enhanced membrane penetration had the opposite effect [17, 21] and also increased the apparent Ca$^{2+}$ affinity of neurotransmitter release at hippocampal autapses [44]. These findings indicate that the ability of the loop 1 and 3 hydrophobic residues to insert into the membrane is crucial to synaptotagmin's role in driving vesicle fusion.

Interestingly, each C2 domain seemed to play equal roles *in vitro*, with C$_2$B even depending on the presence of C$_2$A to insert into membranes or bind liposomes [16]. Furthermore, in isolated C$_2$A domains, single residue substitutions of the loop 1 or loop 3 hydrophobic residues that blocked membrane penetration had summative effects in decreasing the Ca$^{2+}$ dependence of liposome binding, while the double mutant effectively prevented liposome binding [35]. Thus, *in vitro* evidence suggested balanced roles for the C$_2$A and C$_2$B domains with regard to membrane penetration. Our current finding, that the hydrophobic residues in both domains are essential for synaptotagmin-triggered neurotransmitter release, is more consistent with the *in vitro* studies. However, disruption of C$_2$B function *in vivo* is still more severe than that of C$_2$A: 1) mutation of the loop 3 hydrophobic residue in C$_2$B results in dominant negative lethality, while mutation of both the loop 1 and loop 3 hydrophobic residues in C$_2$A does not ([19], and this study); and 2) mutations of Ca$^{2+}$-binding residues in C$_2$B also result in dominant negative phenotypes, while those in C$_2$A do not [2, 10, 29]. The predominance of the C$_2$B domain for synaptotagmin function *in vivo* could be due to its direct interactions with the SNARE complex [21, 31, 37, 40] or due to its greater ability to induce membrane bending [17, 18, 21].

For both C$_2$A and C$_2$B, mutation of the hydrophobic residues required for Ca$^{2+}$-dependent membrane penetration resulted in a more severe phenotype than mutation of the Ca$^{2+}$ binding residues themselves [2, 10, 29]. This combination of findings indicates that: 1) the hydrophobic residues mediate a key effector interaction(s), and 2) the reported mutations of Ca$^{2+}$ binding residues must not completely block these downstream interactions. For example, the *syt*$^{A-ME,FE}$ hydrophobic mutation abolished all synaptotagmin-triggered neurotransmitter release (Fig 2D). Yet the mutation of the second of the five Ca$^{2+}$ binding aspartates to a glutamate (*syt*$^{A-D2E}$), which completely blocked Ca$^{2+}$ binding by C$_2$A *in vitro*, supported some (20%) synaptotagmin-triggered Ca$^{2+}$-evoked release *in vivo* [10]. This combination of effects suggests

that in the *P[sytA-D2E]* mutant, the intact C$_2$B domain may be able to place the C$_2$A pocket close enough to membranes for some hydrophobic interactions to occur, and facilitate a small amount of synaptotagmin-triggered fusion, despite the electrostatic repulsion by C$_2$A. While in the *P[syt$^{A-ME}$,FE]* mutant, the required downstream hydrophobic interactions by C$_2$A are completely blocked and no synaptotagmin-triggered release occurs in response to Ca$^{2+}$ influx. For C$_2$B, the loop 3 hydrophobic mutant resulted in less Ca$^{2+}$-evoked release than seen in *syt$^{null}$* mutants, no change in spontaneous release, and embryonic lethality ([19] and unpublished observations). On the other hand, mutation of Ca$^{2+}$ binding aspartates in C$_2$B to neutral asparagines (*syt$^{B-DN}$*) resulted in less evoked release than in *syt$^{null}$* mutants, but some could survive to larval or even adult stages, and there was an increase in spontaneous transmitter release [2]. No synaptotagmin-triggered, Ca$^{2+}$-evoked release was possible in either case. However, the decreased electrostatic repulsion of the presynaptic membrane in the *P[sytB-DN]* mutants could have permitted membrane interactions by the hydrophobic residues resulting in increased spontaneous neurotransmitter release. Since these mutants were viable to a later stage than the loop 3 hydrophobic mutants, this combination of effects suggests that the increase in spontaneous release in the *P[sytB-DN]* mutants was beneficial to the organism. Thus, the increased severity of the hydrophobic mutations in both C2 domains, compared to the Ca$^{2+}$ binding mutations reported to date, suggests that the interactions mediated by the hydrophobic tip residues are absolutely essential for synaptotagmin function, are downstream of Ca$^{2+}$ binding, and still occur to some degree in Ca$^{2+}$ binding mutants.

Our results, coupled with previous *in vitro* and *in vivo* findings, are consistent with the idea that membrane insertion by both C2 domains is a primary downstream effector interaction in synaptotagmin's transduction of Ca$^{2+}$ influx into vesicle fusion. Prior to Ca$^{2+}$ influx, the negative charge of both Ca$^{2+}$-binding pockets provides electrostatic repulsion of the negatively-charged presynaptic membrane (Fig 1). After Ca$^{2+}$ entry, Ca$^{2+}$ binding results in a net positive charge at the tip of each C2 domain which then attracts the presynaptic membrane–flipping the electrostatic switch [11, 12]. The hydrophobic residues at the tips of each C2 domain are then able to insert into the hydrophobic core of the presynaptic membrane [14, 20, 45]. This insertion has been shown to result in positive curvature of the target membrane [17, 18] and disruption of phospholipid order [46], which could provide the final energy required to drive fusion of the vesicle membrane with the presynaptic membrane. A certain threshold of lipid destabilization could be required for fusion, and inhibiting penetration by C$_2$A may lower membrane destabilization below this threshold.

A recent study [47] suggests a mechanism whereby the membrane disruption resulting from synaptotagmin penetration could directly promote SNARE-mediated vesicle fusion. Synaptotagmin penetration is proposed to disorder lipids near the transmembrane domain of syntaxin allowing the bent linker in the juxtamembrane region to straighten, thereby facilitating the transition from trans- to cis-SNAREs required for driving fusion. Synaptotagmin-triggered membrane curvature [17, 18] could augment this transition. Assuming a certain threshold of lipid disorder is necessary for SNAREs to alter their conformation, partial disruption of the membrane in our *P[syt$^{A-ME}$]* mutant or in the previously studied *P[sytA-FE]* mutant [19] may allow only a subset of SNARE complexes to fully straighten leading to the observed 50% reduction in vesicle fusion. However, our double mutant (*P[syt$^{A-ME}$,FE]*), which should prevent any insertion by C$_2$A, would not contribute to the membrane disorder necessary for conformational change in the SNARE complex, thereby preventing fusion.

A remaining question is how do synaptotagmin 1 mutants result in less fast, synchronous fusion than that seen in the absence of the wild type protein (*syt$^{null}$*)? In these cases, the presence of the mutant protein must actively inhibit the residual neurotransmitter release remaining in *syt$^{null}$* mutants (Fig 2D and [2, 19]). The finding that the Ca$^{2+}$ dependence of evoked

release in *syt^{null}* mutants is the same as in wild type (no change in EC50 [2]) is consistent with the idea that another isoform of synaptotagmin could be the trigger. However, the only other synaptotagmin isoform expressed at the neuromuscular junction in *Drosophila* is synaptotagmin IV and it is concentrated in the postsynaptic cell [48]. Whether there is any synaptotagmin IV in the presynaptic terminal or if its expression level is impacted in *syt^{null}* mutants is unknown. In short, the Ca$^{2+}$ sensor that triggers the residual vesicle fusion in *syt^{null}* mutants remains to be determined.

Regardless of the mechanism of action, our characterization of the most severe C$_2$A domain mutation to date challenges the predominant model of synaptotagmin function. While all previous *in vivo* evidence suggested that C$_2$A acted only as a facilitatory domain, disrupting the hydrophobicity of the loop 1 and loop 3 Ca$^{2+}$-binding pocket residues of C$_2$A (Fig 1) decreased evoked neurotransmitter release to levels less than that in *syt^{null}* larvae (Fig 2). These findings are consistent with *in vitro* evidence indicating important roles for these hydrophobic tip residues in membrane penetration, and potential interactions with membrane embedded SNARE proteins. Our findings now demonstrate that, in contrast to the current view, the C$_2$A domain is absolutely required for synaptotagmin to trigger fast, synchronous vesicle fusion.

## Supporting information

**S1 Raw images.**
(PDF)

## Acknowledgments

Special thanks to Dr. Ann Hess, Colorado State University, Dept of Statistics.

## Author Contributions

**Conceptualization:** Matthew R. Bowers, Noreen E. Reist.

**Formal analysis:** Matthew R. Bowers.

**Funding acquisition:** Noreen E. Reist.

**Investigation:** Matthew R. Bowers.

**Methodology:** Matthew R. Bowers, Noreen E. Reist.

**Project administration:** Noreen E. Reist.

**Resources:** Noreen E. Reist.

**Supervision:** Noreen E. Reist.

**Visualization:** Matthew R. Bowers.

**Writing – original draft:** Matthew R. Bowers.

**Writing – review & editing:** Matthew R. Bowers, Noreen E. Reist.

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
