## [Decision Letter · Decision Letter 0]

12 Dec 2019

PONE-D-19-32484

The C2A domain of synaptotagmin is an essential component of the calcium sensor for synaptic transmission

PLOS ONE

Dear %Noreen%,

Thank you for submitting your manuscript to PLOS ONE. After careful consideration, we feel that it has merit but does not fully meet PLOS ONE’s publication criteria as it currently stands. Therefore, we invite you to submit a revised version of the manuscript that addresses the points raised during the review process.

All of the requested changes are minor, and once these have been addressed the manuscript should be suitable for publication. Can I also remind you that PLOS ONE requires that submissions reporting blots or gels include original, uncropped blot/gel image data as a supplement or in a public repository. If this is addressed in the revised manuscript it will expediate the review process. 

We would appreciate receiving your revised manuscript by Jan 26 2020 11:59PM. To enhance the reproducibility of your results, we recommend that if applicable you deposit your laboratory protocols in protocols.io, where a protocol can be assigned its own identifier (DOI) such that it can be cited independently in the future. For instructions see: http://journals.plos.org/plosone/s/submission-guidelines#loc-laboratory-protocols

We look forward to receiving your revised manuscript.

Kind regards,

Michael A Cousin, PhD

Academic Editor

PLOS ONE

Journal Requirements:

Reviewers' comments:

Reviewer's Responses to Questions

**Comments to the Author**

1. Is the manuscript technically sound, and do the data support the conclusions?

Reviewer #1: Yes

Reviewer #2: Yes

2. Has the statistical analysis been performed appropriately and rigorously? 

Reviewer #1: Yes

Reviewer #2: Yes

3. Have the authors made all data underlying the findings in their manuscript fully available?

Reviewer #1: Yes

Reviewer #2: Yes

4. Is the manuscript presented in an intelligible fashion and written in standard English?

Reviewer #1: Yes

Reviewer #2: Yes

5. Review Comments to the Author

Reviewer #1: In this report, the authors address a long-debated and topical question concerning the role of synaptotagmin C2A domain in mediating neurotransmitter release. Using electrophysiological recording in Drosophila larval neuromuscular junction, combined with targeted mutations, the authors show that calcium triggered membrane insertion of C2A domain of synaptotagmin is critical for neurotransmitter release. They strikingly find that targeted mutation of hydrophobic residues within the calcium binding loop of the C2A domain results in more severe defect than those seen in synaptotagmin null mutants. The experiments are well designed, technically sound and are appropriately interpreted. I strongly support the publication of this work. I would suggest to make following minor adjustments:

1) The author should include information on the calcium binding mutations in both C2A and C2B domain of synaptotagmin and discuss their result in context of those findings.

2) To the best of knowledge, disrupting calcium binding to C2A domain produced only partial effect as compared to near complete loss of function with comparable C2B mutation. How do the authors reconcile these findings? This should be discussed.

Reviewer #2: Bowers and Reist present work on the role of the C2A domain of synaptotagmin (syt) in the Ca2+ dependent stimulation of exocytosis during neurotransmitter release. While there is consensus on the fact that syt is the calcium sensor for Ca2+ regulated release, no consensus has been found on the mechanism. Previously the role of the two C2 domains C2A, and C2B, had been studied and strong impairment of synaptic transmission upon mutation of the C2B domain had been found. In this work, the authors focus on the C2A domain and more specifically how mutations of the Ca2+ binding sites, responsible for membrane interactions, alter neurotransmitter release. In light of the open questions in the field, this work is timely and relevant.

The authors found that mutation of both binding sites in C2A almost completely abolishes evoked release in the Drosophila neuromuscular junction. This finding changes the role of C2A domain from a more facilitating role to an essential role for the function of syt. This result is very interesting because it shifts the discussion about the mechanism of Ca2+ triggered vesicle fusion to the protein/lipid interaction. As the authors discuss, this is consistent with recent findings in the field.

I would suggest that the authors address the following issue:

During the manuscript the authors refer only to “synaptotagmin”. It is mentioned at one point that it is synaptotagmin 1. Which other isoforms of syt are present in the neuromuscular junction and how could they potentially influence the results? Are the expression levels of other isoforms influenced by the mutations?

6. PLOS authors have the option to publish the peer review history of their article (what does this mean?). If published, this will include your full peer review and any attached files.

Reviewer #1: No

Reviewer #2: No

---

## [Author Response · Author response to Decision Letter 0]

10 Jan 2020

Thank you for the opportunity to resubmit a revised draft of our manuscript, “The C2A domain of synaptotagmin is an essential component of the calcium sensor for synaptic transmission.” We are grateful to the reviewers for their valuable feedback and that they “strongly support the publication of this work” and find it to be “timely and relevant.” We have addressed reviewer comments, as summarized below.

Comments from Reviewer 1:

Major Comment 1: “The author should include information on the calcium binding mutations in both C2A and C2B domain of synaptotagmin and discuss their result in context of those findings.”

Response: We have expanded our discussion of Ca2+ binding mutations in the C2A and C2B domains and the comparison to the hydrophobic mutations. (Lines 497-505)

Major Comment 2: “To the best of knowledge, disrupting calcium binding to C2A domain produced only partial effect as compared to near complete loss of function with comparable C2B mutation. How do the authors reconcile these findings? This should be discussed.”

Response: Thank you for pointing out that our discussion regarding the relative role of C2A vs. C2B was not clear enough. There is indeed a difference in neurotransmitter release depending on which domain has been mutated – C2B is still the predominant domain. We have expanded our discussion section to clarify this important point. (Lines 451-453, 507-517)

Comments from Reviewer 2:

Major Comment 1: “During the manuscript the authors refer only to “synaptotagmin”. It is mentioned at one point that it is synaptotagmin 1. Which other isoforms of syt are present in the neuromuscular junction and how could they potentially influence the results? Are the expression levels of other isoforms influenced by the mutations?”

Response: The reviewer brings up an important point that other synaptotagmin isoforms could impact synaptotagmin 1 mutations. In Drosophila, the only syt genes expressed at the neuromuscular junction are syt 1 and syt IV and syt IV is concentrated in the postsynaptic cell. Unfortunately, the impact of syt I mutations on syt IV expression levels is unknown. We have expanded our discussion section to include this information. (Lines 574-585)

Again, we want to thank the reviewers for their thoughtful feedback, and we believe it has made the manuscript stronger.

---

## [Editor Report · Decision Letter 1]

14 Jan 2020

The C2A domain of synaptotagmin is an essential component of the calcium sensor for synaptic transmission

PONE-D-19-32484R1

Dear Dr. Reist,

We are pleased to inform you that your manuscript has been judged scientifically suitable for publication and will be formally accepted for publication once it complies with all outstanding technical requirements.

With kind regards,

Michael A Cousin, PhD

Academic Editor

PLOS ONE
---

## [Editor Report · Acceptance letter]

23 Jan 2020

PONE-D-19-32484R1 

The C2A domain of synaptotagmin is an essential component of the calcium sensor for synaptic transmission 

Dear Dr. Reist:

I am pleased to inform you that your manuscript has been deemed suitable for publication in PLOS ONE. Congratulations! Your manuscript is now with our production department. 

With kind regards,

on behalf of

Prof. Michael A Cousin 

Academic Editor

PLOS ONE